# Mangiferin-Enriched Mn–Hydroxyapatite Coupled with β-TCP Scaffolds Simultaneously Exhibit Osteogenicity and Anti-Bacterial Efficacy

**DOI:** 10.3390/ma16062206

**Published:** 2023-03-09

**Authors:** Subhasmita Swain, Janardhan Reddy Koduru, Tapash Ranjan Rautray

**Affiliations:** 1Biomaterials and Tissue Regeneration Lab., CETMS, ITER, Siksha ‘O’ Anusandhan (Deemed to Be University), Bhubaneswar 751030, Odisha, India; 2Department of Environmental Engineering, Kwangwoon University, 20 Kwangwoon-Ro, Wolgye-Dong, Nowon-Gu, Seoul 01897, Republic of Korea

**Keywords:** mangiferin, hydroxyapatite, β-TCP, BCP, osteogenicity, anti-bacterial

## Abstract

Biphasic calcium phosphate (BCP) containing β-tricalcium phosphate and manganese (Mn)-substituted hydroxyapatite (HAP) was synthesized. Biomedical scaffolds were prepared using this synthesized powder on a sacrificial polyurethane sponge template after the incorporation of mangiferin (MAN). Mn was substituted at a concentration of 5% and 10% in HAP to examine the efficacy of Mn at various concentrations. The phase analysis of the as-formed BCP scaffold was carried out by X-ray diffraction analysis, while the qualitative observation of morphology and the osteoblast cell differentiation were carried out by scanning electron microscopy and confocal laser scanning microscopy techniques. Gene expressions of osteocalcin, collagen 1, and RUNX2 were carried out using qRT-PCR analyses. Significantly higher (*p* < 0.05) levels of ALP activity were observed with extended osteoblast induction on the mangiferin-incorporated BCP scaffolds. After characterization of the specimens, it was found that the scaffolds with 10% Mn-incorporated BCP with mangiferin showed better osteogenicity and simultaneously the same scaffolds exhibited higher anti-bacterial properties as observed from the bacterial viability test. This study was carried out to evaluate the efficacy of Mn and MAN in BCP for osteogenicity and antibacterial action.

## 1. Introduction

Bone makes up the majority of the connective tissue mass in the body. Bone matrix is physiologically mineralized, unlike most other connective tissue matrices, and it is constantly rebuilt throughout the life of a human being because of the formation of new bones. Bone is a heterogeneous composite material consisting of a mineralized phase called hydroxyapatite (HAP; Ca_10_(PO_4_)_6_(OH)_2_), an organic phase (90% type I collagen), 5% non-collagenous proteins (NCPs), and 2% lipids together with water. Fracture healing is a physiological process that leads to bone union. However, large bone defects, despite surgical stability, does not heal spontaneously and entails additional intervention such as natural bone grafting (with an autograft, allograft, or xenograft) [1,2,3]. However, natural bone grafting has several disadvantages. Hence, synthetic calcium phosphate (CaP) bone grafts have been used in many cases. Its exceptional biocompatibility with bone tissues arises due to its chemical composition, similar to the bone mineral phase. Furthermore, CaP ceramic has nontoxic properties and can be attached to bone directly [4,5].

Tissue engineering (TE) is a broad and transdisciplinary field that has shown considerable promise in developing living substitutes for harvested tissues and organs for transplant and reconstructive surgery. TE relies heavily on materials and fabrication technologies to create temporary, synthetic extracellular matrices that supports formation of 3D tissues. Growing cells in 3D scaffolds have become increasingly popular for engineering tissues of realistic size scale and specified forms. They aid in creating functional tissues and organs by guiding cell growth and synthesizing extracellular matrix and other biological substances [6,7]. There are some primary conditions for building polymer scaffolds that are commonly accepted. The first condition is that it must have sufficient porosity and pore size. Secondly, a large amount of surface area is required. Biodegradability is a primary property of the scaffolds with a breakdown rate corresponding to the pace of new tissue creation. For retaining the tissue structure, the scaffold must possess corresponding mechanical strength to that of natural bones. The scaffold should not exhibit any toxicity to the cells. Finally, there should be a positive interaction between scaffold and tissues, resulting in improved cell adhesion, differentiation, migration, and growth [8].

Bioactive ceramics are recognized as the most promising biomaterials for bone tissue engineering. Because of its potential for direct bone-to-implant interaction, ceramics like hydroxyapatite, bioactive glass, β-tri-calcium phosphate (β-TCP), and calcium silicate have been extensively studied for biomaterial applications [9]. For over three decades, biphasic calcium phosphate (BCP) ceramics, which is a blend of HAP and β-TCP, have been broadly utilized as substitute biomaterials for synthetic bone grafting for which it has gained a lot of interest. BCP is appropriate for artificial bone applications and is considered superior to individual phase HAP or β-TCP components due to its exceptional controlled dissolving properties, which enhance new bone development at the implantation site. The degradation rate of β-TCP is 20 times higher than that of HAP [10]. Due to its high brittleness and low fracture toughness, HAP can only be used in non-load bearing areas in clinical orthopaedic and dental applications. β-TCP has less mechanical strength than HAP [11]. As a result, a mixture of HAP and β-TCP would balance out each other’s shortcomings. Thorough characterization of BCP is very important because it offers a combination of improved mechanical stability and bioactivity, which is challenging to accomplish in single-phase materials [12].

Adding trace metal elements (Ag^+^, K^+^, Na^+^, Sr^2+^, Zn^2+^, Cu^2+^, Mn^2+^, Mg^2+^, Al^3+^, Fe^3+^, Th^4+^) significantly improves the physical and chemical properties of bioceramics. The HAP phase contains a large number of trace metal elements [13]. Manganese (Mn) may be added to BCP as a sintering additive to enhance the mechanical characteristics of the material. In the presence of Mn, ligand affinity rises, resulting in increased cell adhesion. Mn in the bones has been observed to reduce bone resorption, according to one study [14]. Mn was also found to act as a calcination and sintering additive in BCP powders without causing establishment of any other subordinate phases such as α-TCP and CaO. Manganese doping in BCP is expected to increase the physicochemical characteristics of the material, resulting in better biological function in antimicrobial efficacy [15].

Moreover, Mn also has a role in the production of mucopolysaccharides, which are necessary for cartilage development [16]. Apart from osteogenicity, Mn^2+^ macrocyclic complexes have many biological properties. Various bacterial strains were utilized in the antibacterial test on Mn. It was observed that Mn-BCP showed outstanding antibacterial action against all of the bacteria examined. Gram-positive and Gram-negative pathogens were both strongly suppressed by MnBCP [17].

Mangiferin (MAN; 2-D-glucopyranosyl-1,3,6,7-tetrahydroxy-9H-xanthan-9-one), a naturally occurring polyphenol found in mango and papaya, is a natural immunomodulator. MAN is antiallergic, antidiabetic, antibacterial, antioxidant, immunomodulatory, and hypocholesterolemic and has some other health-promoting characteristics [18]. MAN also boosts the monocyte–macrophage system capacity and possesses an antibacterial effect against both Gram-positive and Gram-negative pathogens. MAN may be a viable alternative therapy for treating osteolytic bone disorders due to its anti-NF-κβ characteristics. MAN has also been shown to suppress bone apoptosis and the production of osteoclasts. MAN boosted the growth of human bone formation cells considerably, and there was no evidence of cytotoxicity. Moreover, MAN could stimulate the production of alkaline phosphate (ALP) inside human osteoblast cells [19,20].

The synthesis of 5% and 10% Mn-doped BCP-MAN scaffolds has been carried out in this investigation and the efficacy of mangiferin and manganese in terms of enhanced bone regeneration and antibacterial action has been described.

## 2. Materials and Methods

Polyvinyl alcohol (PVA) was obtained from Labo Chemie Pvt. Ltd., Mumbai, India (degree of polymerization ~1700–1800 and hydrolysis ~98–99%). Diammonium hydrogen phosphate ((NH_4_)_2_HPO_4_), calcium nitrate tetrahydrate (Ca(NO_3_)_2_.4H_2_O), manganese(II) chloride tetrahydrate (MnCl_2_.4H_2_O), and mangiferin (C_19_H_18_O_11_) were obtained from Merck Specialities Pvt. Ltd. (Mumbai, India), Sisco research laboratories Pvt. Ltd. (Mumbai, India), India and Himedia laboratories Pvt. Ltd. (Thane, India), respectively. Ammonia was obtained from the Sisco research laboratories Pvt. Ltd., India. All chemicals utilized in this study were of analytical grade.

### 2.1. Fabrication of Mn-BCP Porous Scaffolds

The aqueous precipitation technique was adopted to fabricate 5% and 10% Mn-doped BCP scaffolds. Measured quantities of Ca(NO_3_)_2_4H_2_O and MnCl_2_.4H_2_O were added dropwise to a (NH_4_)_2_HPO_4_ solution at room temperature maintaining a pH of 11. This white coloured product was thoroughly washed with deionized water and aged for 24 h and the solution was then filtered [21]. The resultant product was thermally processed for two hours at 1000 °C to yield Mn-BCP containing Mn-HAP and β-TCP. Next, the as-formed Mn-BCP powder was ground using a mortar and pestle and thereafter sieved to produce particles with a size less than 75 μm. For preparation of the scaffolds, fully reticulated polyurethane (PU) sponge was employed as a sacrificial template. Surface treatment of the sponge was carried out using a NaOH solution for about half an hour to increase its hydrophilicity, and templates of the PU sponges were carved into proper dimensions. PVA was mixed with water at a concentration of 0.1 mol/L to make a slurry. The 5% and 10% Mn-doped BCP powders were further mixed with the slurry in a 30:70 ratio by weight. PU sponges were cleaned and dried before being soaked into the Mn-doped BCP slurry and after uniform soaking, the PU sponges were lightly squeezed to remove excess slurry, and the leftover slurry was then blown with compressed air to achieve a uniform dispersion throughout the sponge. Before getting fired in an electric furnace (at 1200 °C for two hours), the MnBCP-coated sponges were dried at 37 °C for 48 h and then cooled at a rate of 5 °C/min until the temperature reached 25 °C [22,23]. The 5% and 10% Mn-doped BCP scaffolds were soaked in 1 μg/mL mangiferin mixed with dimethylsulfoxide (DMSO) and kept at 37 °C until completely dried.

### 2.2. Characterization of the Scaffolds

#### 2.2.1. X-ray Diffraction

To carry out the phase characterization of the synthesized 5% and 10% Mn-doped BCP scaffolds, an X-ray diffractometer using Cu Kα radiation (λ = 0.154 nm) operated at 40 kV and 20 mA was employed for qualitative analysis. A scan speed of 2° per minute in the range of 10° ≤ 2θ ≤ 80° was used to record the XRD patterns.

#### 2.2.2. Contact Angle Measurement

The wettability of scaffolds was determined using Dulbecco’s modified Eagle’s medium (DMEM) cell culture media and simulated body fluid (SBF). Static contact angles of the immobilized liquid drops were assessed utilizing contact angle equipment at pH 7.2.

#### 2.2.3. Water Uptake Capability

The water absorption capability of the prepared scaffolds was tested by immersing a measured amount of the scaffold in distilled water for 2 h. Thereafter, the scaffolds were removed from the water, the excess water was removed, and their wet weight was calculated [24]. The following formula was used to determine the extent of water absorption: Water uptake capability (%) = Wet weight − dry weight /dry weight × 100

#### 2.2.4. Mechanical Property Measurement

The mechanical properties of the scaffolds were measured at 3, 5, 7, 9, and 11 weeks after degradation and week 0 was used as the base of comparison. The degradation media was replaced every week throughout the degradation period. With the help of an electromechanical universal testing machine (SANSCMT4503, SANS, Shenzhen, China), the compressive strength was evaluated by crushing a 10 *×* 10 *×* 10 mm^3^ scaffold amid two flat platens possessing a ramp rate of 0.5 mm min^−1^. The compressive strength and modulus of scaffold yield were noted for comparison [25].

#### 2.2.5. Biodegradation in SBF

By soaking the scaffolds in SBF at 37 °C, the biodegradability of the scaffolds was examined in vitro. At a solid/liquid ratio of 50 mg/mL, cylinder-shaped scaffolds were soaked in SBF for 1, 5, 10, 15, 20, and 25 days at 37 °C. All the samples were kept in a sealed plastic flask to prevent pH changes and microbial contamination. Throughout the experiment, the SBF solution was not refreshed. The immersed samples were then filtered, rinsed with deionized water, and dried at 40 °C for about four days before being weighed. The percentage of original weight was utilized to compute the weight loss. The weight loss and difference in pH were measured from five scaffolds, and the findings were reported as mean *±* SD [22].

#### 2.2.6. Release of Mangiferin during In Vitro Degradation

In the process of degradation, the scaffolds were taken for characterization at 0, 1, 2, 4, 6, 8, 10, 12, and 14 weeks, and the quantification of MAN release from the scaffolds was performed in vitro. After adding equal volumes of DMSO, MAN was extracted from the scaffold and subsequently centrifuged. High-performance liquid chromatography (Beckman, Brea, CA, USA) was employed for detection of MAN concentration [4].

#### 2.2.7. Ion Release

To determine the ion release characteristics of 5% and 10% Mn-BCP-MAN scaffolds, 500 mg of the sample was immersed in 50 mL SBF. Particle-induced X-ray Emission (PIXE) was utilized to identify the increase of Ca^2+^ as well as Mn^2+^ in the body fluid over time [26].

#### 2.2.8. In Vitro Toxicity Testing Using MTT Assay

In this investigation, the human osteoblast MG63 cell line (obtained from NCCS, PUNE) was used. These cells were incubated at 37 °C in a dehumidified environment containing 5% CO_2_. DMEM (Invitrogen, Paisley, UK) was used to culture the cells, which was supplemented with 10% foetal bovine serum (FBS, Invitrogen, Paisley, UK), 100 mg/mL streptomycin, and 100 U/mL penicillin. Every other day, the cultured media was changed. The MTT (3-[4,5-dimethylthiazol-2-yl]-2,5-diphenyltetrazolium bromide) assay was used to quantify cell proliferation by detecting mitochondrial succinate dehydrogenase function and was utilized to investigate the cytotoxicity of the synthesized scaffolds. The obtained scaffolds were then fixed in the bottoms of 96-well cell culture plates and sterilized for 24 h at room temperature with ethylene oxide (ETO) steam; then, 1 mL of cell suspension was seeded uniformly on each sample. Every two days, the cultured medium was replaced with new medium. Following seeding for 1, 7, and 14 days, 100 mL of MTT (5 mg/mL) solution was added to each well and the detailed procedure was performed as per our previous work [27]. Measurements of four test runs were initiated to evaluate the mean value. The data were evaluated statistically to determine the mean and standard deviation (SD). 

#### 2.2.9. Microscopic Observation and Immunostaining 

The microscopic view of the scaffolds was performed using scanning electron microscopy (SEM) for qualitative analysis of the osteoblast cells along with determination of the pore dimensions of the scaffolds. For assessments using CLSM, colonized cells present on the scaffolds were fixed using 3.7% paraformaldehyde for 20 min. Cell cytoskeletal filamentous actin (F-actin) was visualized by Alexa Fluor 488 Phalloidin (1:25 dilution in PBS, 1.5 h) treatment of cells and counter-staining with propidium iodide (1 μg mL^−1^, 20 min) for labelling of cell nuclei. The cultures were then placed in Vectashield and assessed using a Leica SP2 AOBS (Leica Microsystems, Wetzlar, Germany) microscope [28].

#### 2.2.10. Osteogenic Gene Expression

To measure mRNA gene expression, quantitative reverse transcription-polymerase chain reaction (qRT-PCR) was utilised to analyse the osteogenic differentiation of MG63 cells on scaffold surfaces. Runt-related transcription factor X2 (RUNX2), osteocalcin (OCN), and type-1 collagen were computed using Bio-rad MyiQ2. Cells were cultured at a density of 4 × 10^4^ per well for 1, 7, and 14 days before being lysed with TRIZOI (Invitrogen, Waltham, MA, USA) to obtain RNA. To acquire enough RNA, cells from all scaffolds in each group were used. A total of 1 mg of RNA was reverse transcribed to complementary DNA (cDNA) using the superscript II first-strand cDNA synthesis kit [29].

#### 2.2.11. Alkaline Phosphatase (ALP) Assay

Osteoblast cell differentiation was estimated by ALP activity. Osteoblast cells were lysed in a buffer solution containing 0.05% Triton X-100, 1.0% Tris, and 6.0% NaCl (*w*/*v* in deionized water, pH 10.0. All the chemicals were procured from Sigma Aldrich (St. Louis, MI, USA). A volume of 60 µL of scaffold specimen solution was added to 50 µL of 0.07% p-nitrophenylphosphate (*w*/*v*, Thermo Fisher, Waltham, MA, USA) in amino methyl propanol (AMP, Acros Organics, Pittsburgh, PA, USA) buffer and then the resulting solution was placed in an incubator for 2 h at 37 °C. The absorbance was recorded at 400 nm. ALP activities were normalized to the ALP activity/µg of the entire DNA content [27].

#### 2.2.12. Bacterial Viability Test

The antibacterial study of the 5% and 10% Mn-doped BCP-MAN scaffolds was carried out using *Staphylococcus aureus* (*S. aureus*). Mueller–Hinton Broth (MHB) medium was used to culture the bacteria. The MTT assay was utilized to assess bacterial viability in vitro. The detailed procedures were followed as per our previous work [30]. 

### 2.3. Statistical Analysis

The SPSS (V 22) statistical analysis software was used to perform statistical analyses on the collected data. The mean ± standard deviation was used to express all of the experimental outcomes except XRD, SEM, and CLSM. One-way analysis of variance (ANOVA) was used to compare the changes in the data. A *p* value < 0.05 was considered statistically significant in all the studies.

## 3. Results

### 3.1. X-ray Diffraction

The XRD pattern of 5% Mn-BCP and 10% Mn-BCP are shown in Figure 1. In the synthesized Mn-BCP scaffolds, Mn-HA and β-TCP peaks were present and are quite similar to the stoichiometric HA diffraction peaks (JCPDS no. 9-0432) and (JCPDS no. 9-0169), respectively. The doped Mn-BCP nanoparticles did not show any additional impurities, according to the XRD pattern. 

### 3.2. Contact Angle Measurement

In both media, the 10% Mn-BCP-MAN scaffold had lower contact angles than the 5% Mn-BCP-MAN scaffold (Figure 2). Further investigation revealed that DMEM contact angles of the 5% Mn-BCP-MAN scaffolds were as low as 42° ± 2.04, whereas SBF contact angles were 48° ± 2.44. Meanwhile, DMEM and SBF had contact angles of 35° ± 1.81 and 41° ± 2.06 for 10% Mn-BCP-MAN, respectively.

### 3.3. Swelling Ratio

The water uptake capability of the soaked scaffolds after different soaking periods of 0, 1, 4, and 7 days is depicted in Figure 3. A significant change in swelling ratios of the scaffolds was observed. The swelling was higher in 5% Mn-BCP-MAN scaffolds (287 ± 14), but it decreased to 253 ± 13% for 10% Mn-BCP-MAN scaffolds. 

### 3.4. Mechanical Property of the Scaffold

Compressive strength (Figure 4) and modulus (Figure 5) of the scaffolds were estimated from week 0–11 after degradation. The in vitro compressive strength and modulus at yield considerably decreased in the BCP scaffolds with time. However, BCP-MAN scaffolds with different Mn concentrations did not exhibit any change in the above tests.

### 3.5. Degradation

It can be seen that the rate of degradation of both the scaffolds increased as the soaking period increased (Figure 6). The 10% Mn-BCP-MAN scaffold show a lower deterioration rate in comparison to 5% Mn-BCP-MAN scaffolds. Within the time period of one week, 10% Mn-doped BCP scaffolds lose less weight than 5% Mn-doped BCP scaffolds. The biodegradation ratio reduced as the concentration of Mn increased from 5% to 10%. 

### 3.6. In Vitro Release of Mangiferin from Scaffolds

The release of MAN was closely linked with the concentration of Mn present in 5% and 10% Mn-BCP-MAN scaffolds. The total release of MAN was estimated to be about 90% for each sample (Figure 7). MAN exhibited a fast release from Mn-BCP-MAN (5% and 10%) scaffolds in the 1st week and steadied within 2 weeks of release. There seemed to be little difference between 5% Mn-BCP-MAN and 10% Mn-BCP-MAN scaffolds as far as MAN was concerned. The amount of MAN released from 5% Mn-BCP-MAN (MAN: 96 µg/10 mg) and 10% Mn-BCP-MAN (MAN: 93 µg/10 mg) scaffold in the degradation media were similar at the end of two weeks of degradation. In vitro release of MAN was detected after the 14th week of degradation.

### 3.7. Ion Release

The antibacterial property of the scaffold surface can be revitalized over time by the bioactive Mn ions released by the scaffolds. The ion release behaviour of the 5% and 10% Mn-BCP-MAN scaffolds were studied by immersing the scaffolds in SBF at 37 °C, and ion release was measured using PIXE at various time points. Mn ions were liberated from the scaffolds after 12 h at a concentration of 64 ppm and 79 ppm from 5% Mn-BCP-MAN and 10% Mn-BCP-MAN scaffolds, respectively. The liberated Mn ions play an important role in enhancing the antibacterial properties of Mn-doped BCP-MAN scaffolds.

### 3.8. MTT Assay

The MTT assay was used to examine MG63 cell viability on 5% and 10% Mn-BCP-MAN scaffolds. The cell density of both the scaffolds was evaluated after culturing for 1, 7, and 14 days, as shown in Figure 8. Pure BCP was used as the control specimen. On day 1, cell viability of the 10% Mn-BCP-MAN scaffold was modest, but on days 7 and 14, the cell proliferation rate was higher than the 5% Mn-BCP-MAN scaffold. For all cultured days, the statistical analysis resulted in a significant difference (*p* < 0.05) in cell density between the 5% and 10% Mn-doped BCP-MAN scaffolds. 

### 3.9. SEM and CLSM Observation

Assessment of the 10% Mn-BCP-MAN scaffold by SEM (Figure 9a) and CLSM (Figure 9b) on the 14th day of culture exhibited elongated cells spread throughout the scaffold surface by establishing cell-to-cell contacts on 10% Mn-BCP-MAN. Adherent cells seemed to be well spread with elevated cytoplasmic volume and higher amounts of fibrillar projections. Moreover, the cells showed a well-aligned F-actin cytoskeleton having intense staining at the boundaries of cells with the appearance of prominent nuclei and cell division [31]. 

### 3.10. Osteogenic Gene Expression

Osteogenic gene expression was used to assess the differentiation of MG63 cells on both 5% and 10% Mn-BCP-MAN scaffolds. Pure BCP was used as the control specimen. The expression level of osteogenic genes of COL1A1 (Figure 10), RUNX2 (Figure 11), and OCN (Figure 12) increased from day 1 to day 14 for MG63 cells on both 5% and 10% Mn-BCP-MAN scaffolds. In comparison to the 5% Mn-BCP-MAN scaffold, the 10% Mn-BCP-MAN scaffold demonstrated greater gene expression levels (*p* < 0.05). Table 1 shows the forward and reverse primers for quantification of expression of the relevant genes.

### 3.11. ALP Activity

Measurement of ALP activity was carried out to assess the capability of the scaffolds to accelerate osteoblast cell differentiation (Figure 13). Pure BCP was used as the control specimen. After 7 days of culture, osteoblast cells in contact with the 5% Mn-BCP-MAN scaffold surface exhibited insignificant ALP activity compared to those on the 10% Mn-BCP-MAN scaffolds (*p* < 0.05). However, after 14 days of culture, there seemed to be significantly higher ALP activity on 10% Mn-BCP-MAN scaffolds than the 5% Mn-BCP-MAN scaffolds. 

### 3.12. Bacterial Viability

At 490 nm, optical density measurements were used to examine the activity of *S. aureus*. The data were compiled in 10 h intervals until 30 h. Bacteria growth was monitored on pure BCP control scaffold, 5% Mn-BCP-MAN scaffold, and 10% Mn-BCP-MAN scaffold. During the initial 10 h, there was an insignificant decrease in bacterial cells, but bacterial count exponentially decreased as the time period increased from 20 h to 30 h. In the 10% Mn-BCP-MAN scaffold, there was a significant decrease in bacteria cell count, as illustrated in Figure 14.

## 4. Discussion

Mn-BCP-MAN comprises balanced combinations of a non-resorbable phase (Mn-HAP) and resorbable phase (β-TCP) that frequently demonstrate increased bioactivity, and satisfactory antibacterial properties together with good mechanical strength, which cannot be achieved by a single-phase biomaterial. The diverse action of naturally occurring MAN at the cellular as well as molecular level offers vital knowledge for its usage as a potential osteoporotic agent. It has been established that MAN suppresses the formation of bone resorption cells by inhibiting RANKL-induced activation of NF-kβ and ERK I ligand. Moreover, it enhances the development of bone formation cells by raising OCN, COL1A1, and RUNX2 expression levels [31].

The effectiveness of a sustained-release MAN scaffold in the treatment of diabetic alveolar bone defects was analysed in an earlier study. The resulting scaffolds exhibited porous architectures, possessing pores 111.35 to 169.45 μm in size. Average pore size decreased with increasing PLGA content. Increased drug content was produced by either a decrease in PLGA concentration or an increase in MAN concentration [32]. In in vitro models, the MAN-loaded scaffolds prevented the decline in cell viability due to diabetes. Additionally, healing of delayed alveolar bone defects was improved with enhanced bone regeneration in diabetic mice. Another study was carried out to find out whether treatment of MC3T3-E1 cells with MAN could protect the cells against dexamethasone-induced toxicity. The outcomes showed that incorporation of MAN greatly reduced the effects of dexamethasone on cell viability of MC3T3-E1 cells and levels of ALP activity. Increased OCN is a characteristic of osteogenic differentiation, and ALP activity is regarded as an early marker of this differentiation [33]. In summary, it can be concluded that MAN could be used to treat significant bone disorders.

Figure 1 depicts the XRD pattern from which it was evident that Mn-doped BCP did not show any impurities because of the absence of additional diffraction peaks. The 10% Mn-BCP scaffold showed higher crystallinity as compared to its 5% Mn-BCP counterpart. Although 5% Mn-BCP showed a further increase in the intensity of the β-TCP peak while decreasing the intensity of the HAP peak, the highest amount of β-TCP was detected in this specimen as compared to its 10% Mn-BCP counterpart. These phenomena could be explained by the Mn solubility limit in HAP. The 5% Mn-BCP likely resulted in the production of β-TCP in terms of the second phase; however, a concentration of Mn > 5 mol%, stabilizes the HAP phase, thereby preventing the formation of the subordinate phase. Furthermore, the decrease in the β-TCP peak leads to an enhancement of the HAP peak as the concentration of manganese increases in BCP [34].

The enhancement in the β-TCP peak intensity of Mn concentration to 5 mol% can be attributed to the incorporation of Mn^2+^ ions at the Ca^2+^ ion site in the β-TCP phase. Calcination of Mn-doped BCP at 1000 ℃ stabilizes its phase structure, and this phenomenon explains the decomposition of the structural phase [16]. The intensity of the β-TCP peak marginally deviated to a higher angle of 2θ with an increase in the concentration of Mn, but no change was observed in the HAP peaks. This finding demonstrated that doping of Mn favours the TCP phase over the HAP phase [35].

On the other hand, 5% Mn-doped BCP-MAN exhibited a large contact angle due to its lower hydrophilicity when compared with 10% Mn-BCP-MAN scaffolds. The wettability of the specimens influences cell proliferation, differentiation as well as cell adhesion on the biomaterial surface. Furthermore, an increase in Mn concentration in the scaffold results in decreased contact angles [36].

The swelling ratio of the scaffold is used to estimate impact on cell activities like cell proliferation, growth, and adhesion [37]. The swelling ratio was found to be >100% for all the synthesized samples, thereby stimulating cell development on the scaffold. However, micro, as well as macro pores were present in the synthesized scaffolds for both the 5% and 10% Mn-BCP-MAN scaffolds. The number of macropores was abundant, showing that water absorption increases with an increase in pore size [38].

The newly generated bone is envisaged to substitute the 5% and 10% Mn-BCP-MAN scaffolds and show better mechanical strength because of the presence of higher amount of Mn in it. An ideal scaffold should have a controlled biodegradation rate, which is related to the bone remodelling speed. Enhanced osseointegration should supply ample mechanical strength for the regeneration. During the resorption process, the mechanical strength of the scaffolds must be retained until the implantation area is totally replaced by the host tissues so that it can resume its structural role [39].

According to the literature [11], the dissolution rate of β-TCP is higher inside the body environment in comparison to HAP. The degradation rate of all our synthesized scaffolds was slow and both the specimens demonstrated a consistent degradation rate, which differs from the literature.

Ca and P, as the major mineral components of HAP, exhibit critical functions in accelerating and retarding osteoblast and osteoclast activities. Both 2–4 mmol (low) and 6–8 mmol (medium) content of Ca^2+^ ions are favourable for osteoblast proliferation, differentiation, and extracellular matrix remineralization. On the other hand, P seems to perform as a subordinate in osteoblast proliferation as well as differentiation [12]. 

The Mn-BCP-MAN scaffold was soaked in SBF at 37 °C, and ion content was measured using the PIXE technique for different time periods to explore its ion release properties. The release of Mn^2+^ ions aids in the stimulation of osteoinductivity along with the antibacterial activities of Mn-BCP-MAN scaffolds [40].

Furthermore, the amounts of β-TCP and Mn-doped HAP in the scaffolds regulate cell viability as well as functionality. In the initial phases of the experiment, according to the MTT assay, the survival rate of MG63 cells suggested its cytotoxic nature. On the 14th day, however, the survival rate of MG63 cells on the 10% Mn-BCP-MAN scaffold was comparatively higher than those on the 5% Mn-BCP-MAN scaffold. Thus, the presence of MAN affects the cell proliferation rate.

CLSM observation exhibited organized cellular activities. This behaviour is appropriate as far as biological activities on the scaffolds is concerned, i.e., since the F actin cytoskeleton, that is highly concentrated below the plasma membrane, gives structural strength and elasticity to the cell that undergoes adaptation to the scaffold structure. Moreover, the F-actin cytoskeleton is a primary candidate in the mechano-transduction mechanism of cells that modulates complex signalling pathways that are mandatory to the next stages of osteoblast proliferation and differentiation [41]

The qRT-PCR technique was carried out in order to investigate the osteogenic gene expression of MG63 cells for COL1A1, RUNX2, and OCN. Throughout proliferation as well as matrix maturation stages of osteoblastic cell development, COL1A1 is considered an early-stage marker. On the 7th and 14th day, the osteogenic gene expression for COL1A1 of osteoblast cells on the 10% Mn-BCP-MAN scaffold was higher than the 5% Mn-BCP-MAN scaffold, indicating that the presence of MAN results in increased proliferation as well as differentiation rate. OCN is regulated via RUNX2 and is a RUNX2 target gene [42]. OCN is termed a late-stage gene marker. The presence of MAN significantly boosts RUNX2 transcriptional activity, as discussed by Peng et al. [43]. However, the results revealed that, in the presence of antimicrobial Mn^2+^ ions, the level of gene expression decreased. Because of the higher antibacterial efficacy of Mn, less bacterial cells were viable in the 10% Mn-BCP-MAN scaffold compared to those in the 5% Mn-BCP-MAN scaffold [44].

Relating to the osteo-inductive property of the scaffolds, the ALP activity of osteoblast cells on both the scaffolds on day 7 showed the least variation. However, ALP activity was significantly enhanced on the 10% Mn-BCP-MAN scaffolds than 5% Mn-BCP-MAN scaffolds on day 14. This change may be based on the progressive release of β-glycerophosphate during scaffold degradation that is widely used to stimulate osteoblast cell-mediated mineralization.

## 5. Conclusions

While 10% Mn-BCP-MAN showed higher hydrophilicity, its swelling ratio was less than 5% Mn-BCP-MAN. The release of mangiferin in both the scaffolds showed insignificant variation that led to osteogenicity of the scaffolds. COL1A1, OCN, and RUNX2 showed higher osteogenicity in the case of the 10% Mn-BCP-MAN scaffold and the antibacterial efficacy of 10% Mn-BCP-MAN scaffold increased with the increase in Mn content.

## Figures and Tables

**Figure 1 materials-16-02206-f001:**
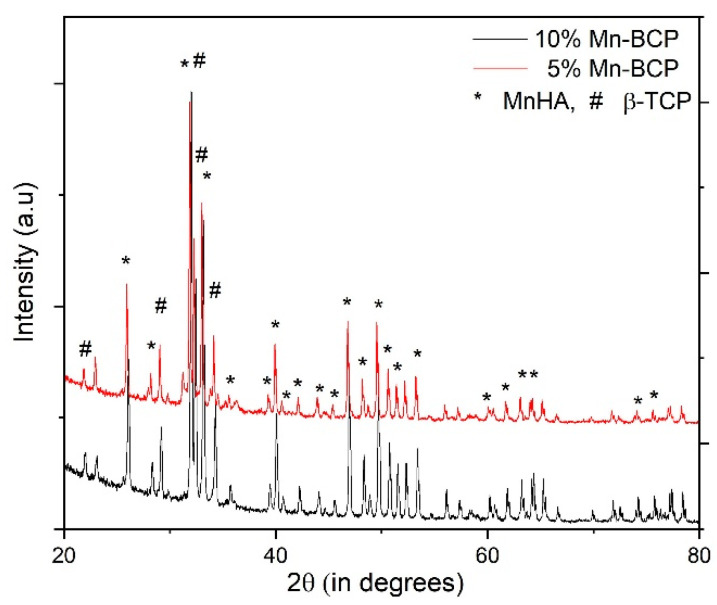
XRD pattern of manganese-doped biphasic calcium phosphate scaffolds.

**Figure 2 materials-16-02206-f002:**
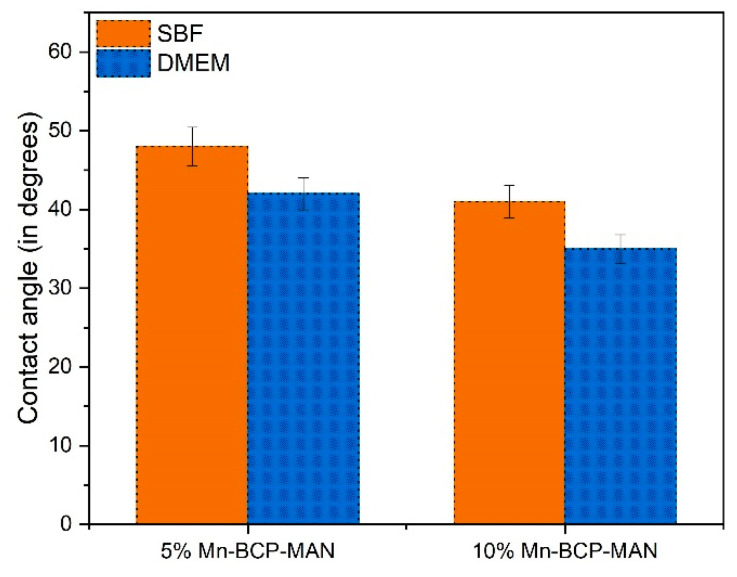
Contact angle of manganese-doped biphasic calcium phosphate scaffolds.

**Figure 3 materials-16-02206-f003:**
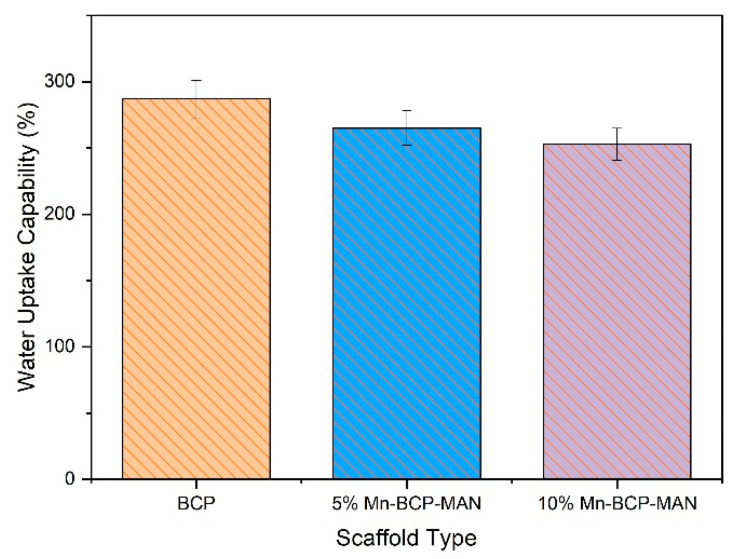
Water uptake capability of the scaffolds.

**Figure 4 materials-16-02206-f004:**
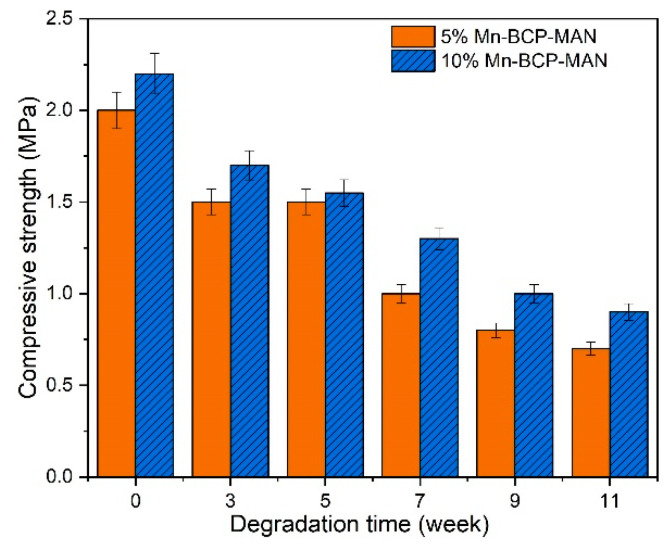
Compressive strength of the scaffolds.

**Figure 5 materials-16-02206-f005:**
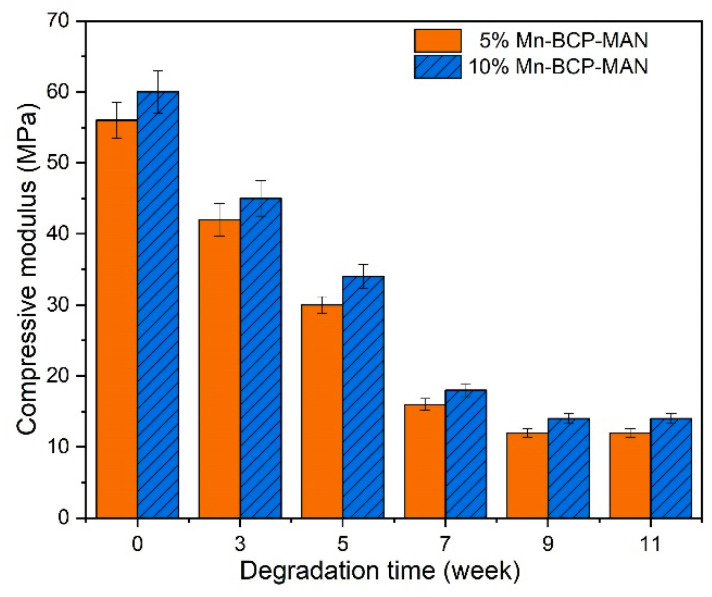
Compressive modulus of the scaffolds.

**Figure 6 materials-16-02206-f006:**
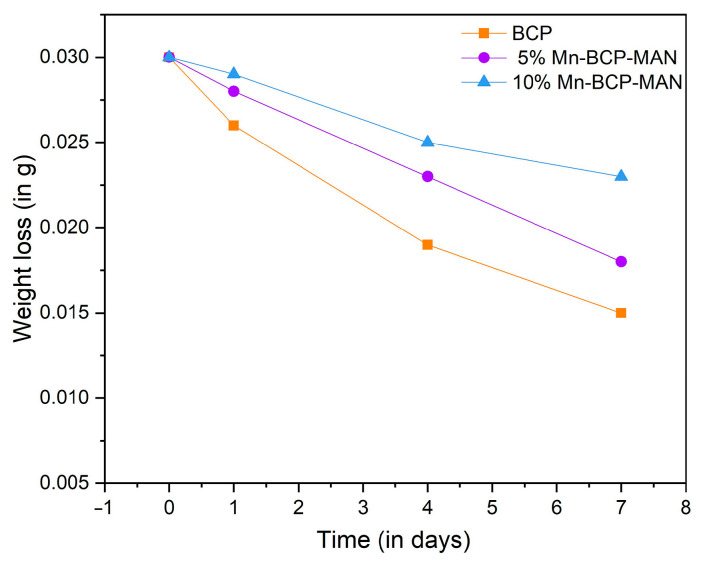
Degradation of the scaffolds for 1-week soaking period.

**Figure 7 materials-16-02206-f007:**
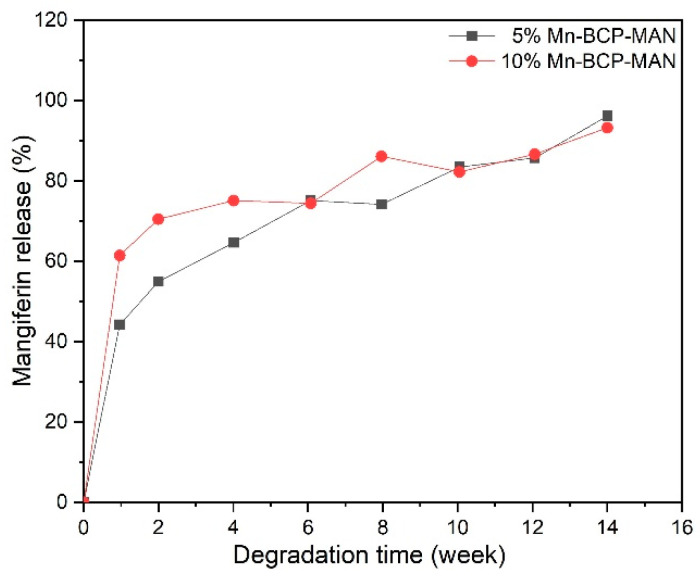
Release of mangiferin from the scaffolds.

**Figure 8 materials-16-02206-f008:**
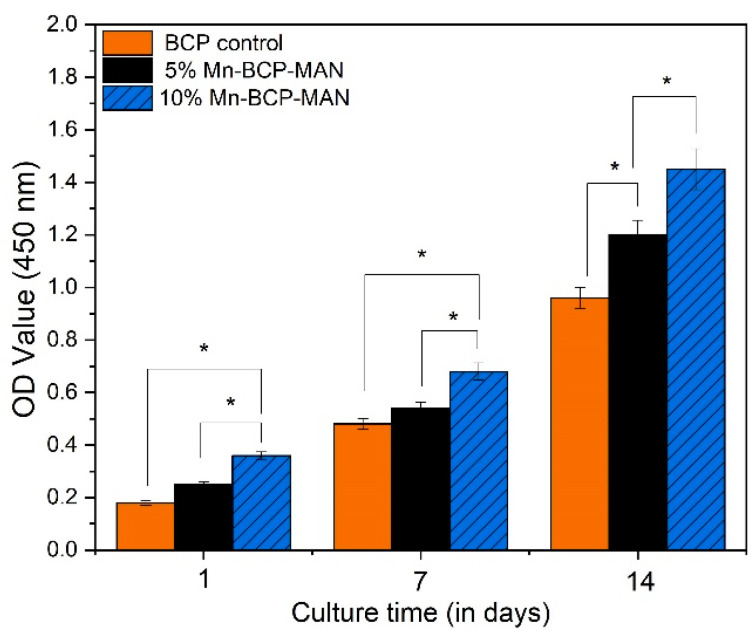
MTT assay of MG63 cells on the scaffolds; * denotes *p* < 0.05.

**Figure 9 materials-16-02206-f009:**
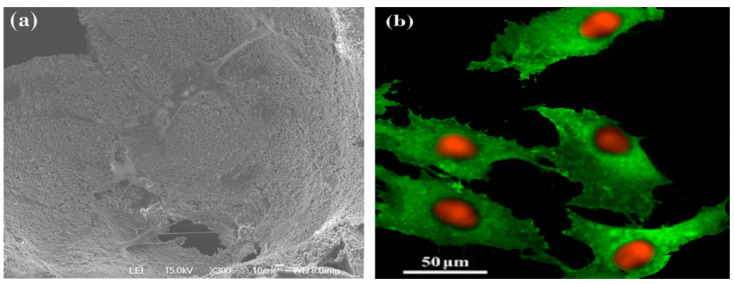
(**a**) Cellular morphology by scanning electron microscopy and, (**b**) confocal laser scanning microscopic image of osteoblasts spread on 10% Mn-BCP-MAN scaffold.

**Figure 10 materials-16-02206-f010:**
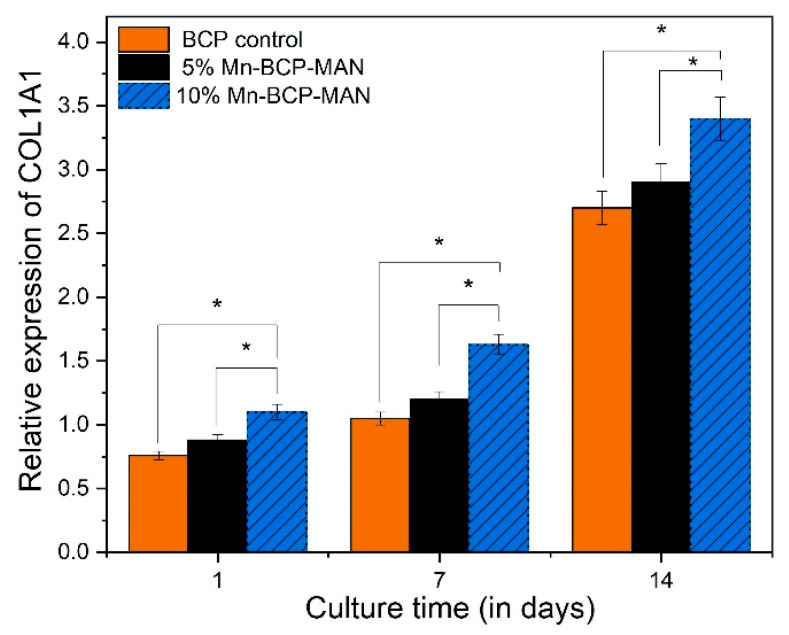
Relative expression of collagen 1; * denotes *p* < 0.05.

**Figure 11 materials-16-02206-f011:**
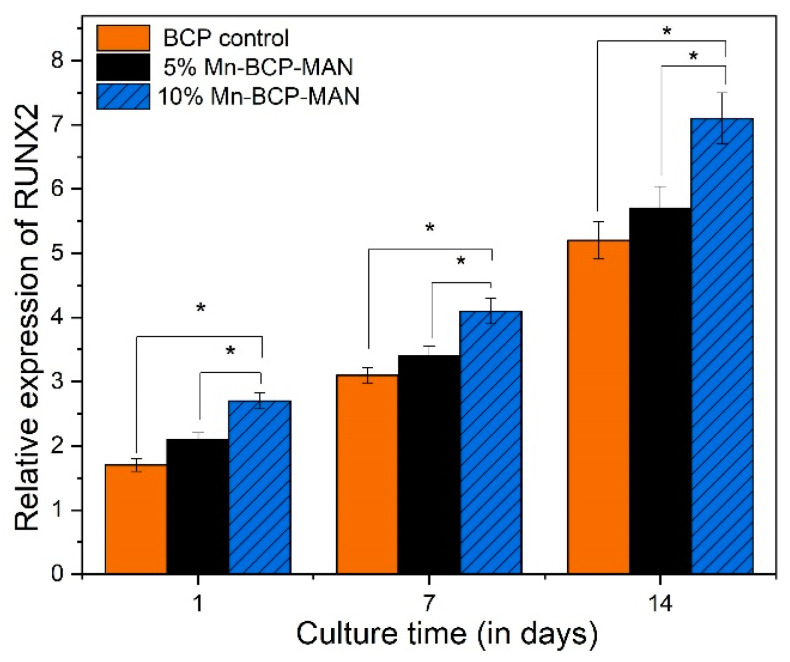
Relative expression of Runt-related transcription factor X2; * denotes *p* < 0.05.

**Figure 12 materials-16-02206-f012:**
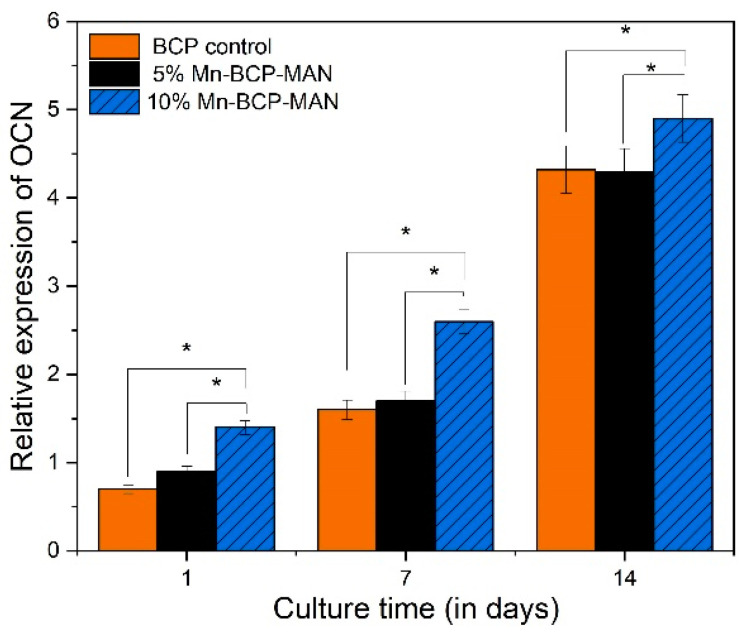
Relative expression of osteocalcin; * denotes *p* < 0.05.

**Figure 13 materials-16-02206-f013:**
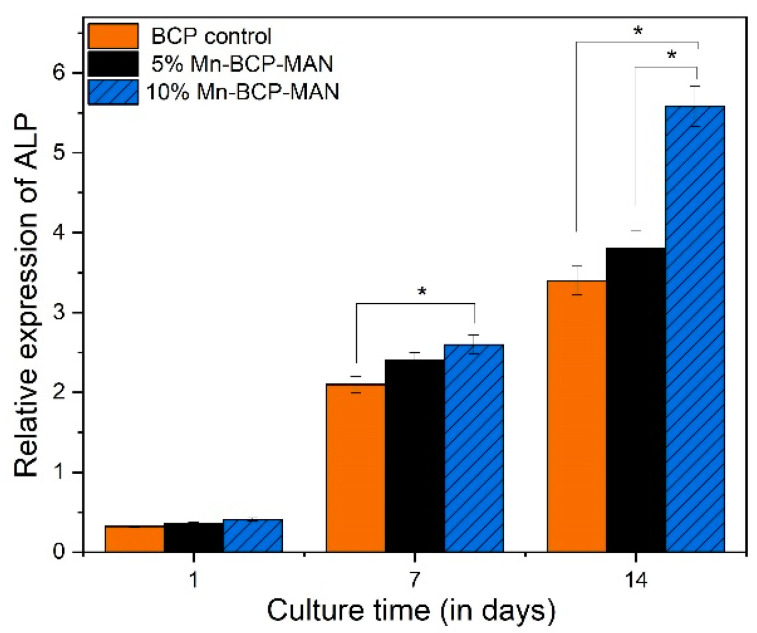
Relative expression of alkaline phosphatase; * denotes *p* < 0.05.

**Figure 14 materials-16-02206-f014:**
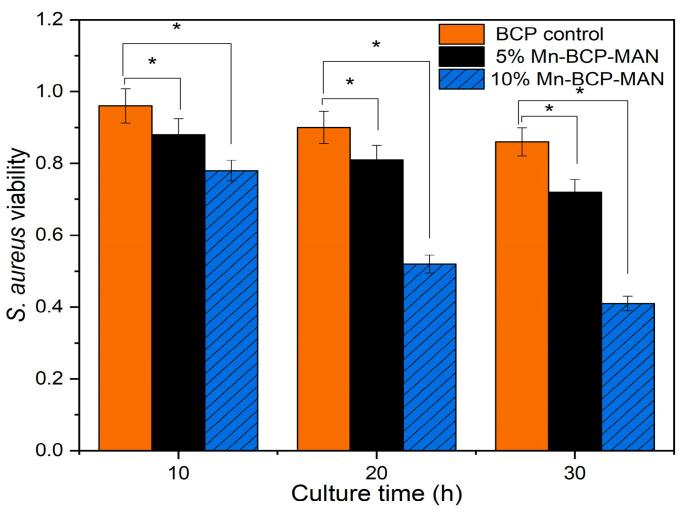
Viability of *Staphylococcus aureus* on the scaffolds; * denotes *p* < 0.05.

**Table 1 materials-16-02206-t001:** Primer sequences.

Gene	Forward Primer Sequence (5′–3′)	Reverse Primer Sequence (5′–3′)
RUNX2	GGATTCTAAGCAAGGCATGG	ATGGGGAAATGTTTGCAATG
COL1A1	CCTCCCCGTGACTGTAGTGT	GAGACCCTGTAGGTGGGAAA
OCN	CTCTGCTCCACAGCCTTTGT	CCTCCTCTCCCTACACATGG

## Data Availability

No new data were created other than reported in this article.

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
