# Peer review of "Mangiferin-Enriched Mn–Hydroxyapatite Coupled with β-TCP Scaffolds Simultaneously Exhibit Osteogenicity and Anti-Bacterial Efficacy"

_materials, 2023, doi:10.3390/ma16062206_

Round 1

Reviewer 1 Report

SUMMARY:

The manuscript entitled “Mangiferin enriched biphasic calcium phosphate (containing 5% and 10% Mn doped hydroxyapatite coupled with β-TCP) favors higher Mn containing scaffolds for osteogenic and anti-bacterial efficacy”, by Subhasmita Swain et al., synthesized biphasic calcium phosphate (BCP) containing β-tricalcium phosphate (β-TCP), manganese (Mn, 5% and 10%), and mangiferin (MAM). The analysis of the scaffolds included the X-ray diffraction, contact angle measurement, water uptake capability, mechanical property measurement, biodegradation in simulated body fluid, release of MAM during in vitro degradation, ion release, in vitro toxicity testing using MTT, microscopic observation and immunostaining, osteogenic gene expression, alkaline phosphatase (ALP) activity, and bacterial viability assays. The authors confirmed that Mn doped BCP, consisting of about 40% Mn-Hydroxyapatite (HA) and 60% β-TCP, was obtained. While 10% Mn-BCP-MAN showed higher hydrophilicity, its swelling ratio was less than 5% Mn-BCP-MAN. The release of MAM in both the scaffolds showed insignificant variation. Also, scaffolds demonstrated osteogenic capacity. Additionally, 10% Mn-BCP-MAN scaffold promoted more COL1A1, OCN, and RUNX2 expression compared to 5% Mn-BCP-MAN. Finally, 10% Mn-BCP-MAN scaffold demonstrated more antibacterial effect than 5% Mn-BCP-MAN scaffold. The subject of the study is relevant.

MAJOR REMARKS:

Title: 

The authors must review the title since it is long and not clear.

Abstract:

The authors stated that the osteogenic differentiation was carried out by scanning electron microscopy and confocal laser scanning microscopy techniques. However, the differentiation was evaluated by alkaline phosphatase activity and gene expression assays. 

The author affirmed that “MG63 cells exhibited more obvious results on the mangiferin incorporated BCP scaffolds with elevated mineralized nodules and significantly higher (p<0.05) levels of ALP activity were observed with extended osteoblast induction”. However, experiments to determine mineralization nodules were not performed.

The antibacterial experiments were not described in the abstract section. However, the authors concluded the scaffold containing 10% manganese promoted higher osteogenic differentiation and antibacterial effect. 

The authors must review and re-write the abstract since it is not clear and is not describing the relevance of the performed study. 

Materials and Methods:

Cellular viability

The authors should have included a cellular positive control, using a reference commercial bone graft material, or at least only cells directly seed on the plaque surface.

Gene expression

A negative and a positive control group should have included. 

Alkaline phosphatase activity

A negative and a positive control group should have included.

Bacterial analysis

The authors evaluated the bacterial effect of the scaffolds only on Staphylococcus aureus. Have not the authors considered evaluating more relevant bacteria in the context of bone regeneration since this is the application of the scaffold?

Additionally, a negative and a positive control group should have included in the antibacterial effect analysis. 

Statistical analysis

The authors must specify which data were included in the statistical analysis. Also, the statistical analysis must be detailed.  

Results:

It is recommended to add the description of all the abbreviated words in the legends of the figures.

The legend of figure 4 is not correct.

“2.6. In vitro release of mangiferin from scaffolds”

The author could re-write this item since the information “In vitro release of MAN was detected after 14th week of degradation” is not the most important information for bone regeneration. Therefore, this information could stay at the end of the paragraph. 

“Fig. 9. (a) SEM morphology and, (b) CLSM image of osteoblasts spread on scaffolds” 

The legend must be reviewed. It is not the SEM morphology. The correct is the cellular morphology according to SEM analysis.

“2.8. MTT assay

(...) On day 1, cell proliferation of the 10% Mn-BCP-MAN scaffold was modest (…)”

MTT assay is used to determine the cell viability and not cell proliferation. 

Discussion

The authors mentioned “During the examination, we discovered that MAN suppresses the formation of bone resorption cells by inhibiting RANKL-induced activation of NF-kβ and ERK I ligand. Moreover, it enhances the development of bone formation cells by raising OCN, COL1A1, and RUNX2 expression levels [21]”. However, the performed experiments were not able to get this information.

MINOR REMARKS:

Abstract: 

“Biphasic calcium phosphate (BCP) containing β-tricalcium phosphate (TCP) and the other component as Mn substituted hydroxyapatite (HAP) was synthesized and its scaffold was fabricated after the incorporation of Mangiferin.” 

It would be appropriate to spell out Manganese (Mn) the first time it appears in the Abstract.

Introduction and Discussion:

Several abbreviations were not spelled out in the Introduction, such as TCP, BCP, and HAP. It is recommended to add the explanation also in these sections. 

Material and Methods

“4.2.11. Alakaline Phosphatase (ALP) assay”

The authors must confirm the spelling of the word "alkaline".

Text sequence:

“2. Results 

2.1. X-ray diffraction”

I recommend the authors confirm if the Materials journal, Advanced Nanomaterials and Nanotechnology Section, accepts the Results section before the Materials and Methods section. 

Grammatical: 

It is recommended the authors revise the manuscript. Readers prefer short and direct sentences. 

Author Response

We are very much thanking the valuable comments of editor and the all reviewers on our revised manuscript. The followings are the explanations presented in reply to each reviewers’ comment. The critical comments and useful suggestions have been helped us to improve our paper considerably. As indicated in the reply’s that follow, we have taken these comments and suggestions into account in the revised version of our manuscript and changes indicates in ‘Track Changed’ format in the revised manuscript. The responses for review comments attached here for your perusal.

Reviewer 2 Report

Regarding 4.1. Fabrication of Mn-BCP composite and Mn-BCP porous scaffolds, the experimental process of 5% or 10% Mn doped hydroxyapatite described by the author is vague. Please clearly describe the preparation process.

 The statistical method is rough about the description of method and results during the manuscript.

 The experimental materials and methods in this paper do not show that the Mn-BCP-MAN scaffold have been prepared. The experimental results on Mn-BCP-MAN scaffold are incredible.

 The experimental design and results lack reasonable explanation and the manuscript is not smooth.

Author Response

We are very much thanking the valuable comments of editor and the all reviewers on our revised manuscript. The followings are the explanations presented in reply to each reviewer’s comment. The critical comments and useful suggestions have been helped us to improve our paper considerably. As indicated in the reply’s that follow, we have taken these comments and suggestions into account in the revised version of our manuscript and changes indicates in ‘Track Changed’ format in the revised manuscript. The responses for review comments are attached here for your perusal.

Reviewer 3 Report

Comments to author

This manuscript entitled “Mangiferin enriched biphasic calcium phosphate (containing 5% and 10% Mn doped hydroxyapatite coupled with β-TCP) favors higher Mn containing scaffolds for osteogen-ic and antibacterial efficacy” reports the efficacy of Mn and Mangiferin for bone regeneration using BCP in vitro. There is some interest, but some modifications are needed.

1.     In Abstract and Introduction, the purpose of this study is not clearly.

2.     In Abstract, abbreviations of “Mn”, “Ca”, and others suddenly appear. I do not think you need to use these abbreviations in abstract if you do not use twice.

3.     In Introduction, I do not think you need to describe “(BG)” and “(CS)” because you do not use these abbreviations since.

4.     In Introduction, the sentence of “Mn doping was done using a wet chemical precipitation approach, followed by heat treatment. The microstructural, physicochemical, and biological features of doped nano-powders were then evaluated and compared. Selective gram-positive pathogenic strain was used to confirm the antibacterial activities. Lastly, its biological properties were validated with the help of osteoblast-like cells by assessing their cytocompatibility and survivability in vitro.“ should be described in Material and Methods. Instead of these, you should indicate clearly the purpose of this study.

5.     Please describe the Material and Methods before the Result.

6.     In Material and Methods, what is PVA?

7.     In Result of X-ray diffraction, I think it is impossible to quantify the HA and beta-TCP.

8.     In Result of swelling ratio, this manuscript describes “swelling ratio of the scaffolds reduced significantly as the Mn concentration increased.” However, no significant difference is visible in figure 3.

9.     In all figures, significant differences should be indicated using some symbol such as asterisk.

10.  In the second paragraph of Discussion, what is Dex?

11.  In the third paragraph of Discussion, this manuscript describes “10% Mn-BCP scaf-fold showed higher crystallinity as compared to its 5% Mn-BCP counterpart. Although, 5% Mn-BCP showed a further increase in the intensity of the β -TCP peak while decreasing the intensity of the HAP peak, the highest amount of β -TCP was detected in this specimen as compared to its 10% Mn-BCP counterpart.” However, since the two lines in Figure 1 overlap, it is difficult to understand. Is it possible to fix it by dividing it into two stages, upper and lower?

12.  In Conclusion, there should also be a concise conclusion.

13.  Please make the font sizes the same.

Author Response

We are very much thanking the valuable comments of editor and the all reviewers on our revised manuscript. The followings are the explanations presented in reply to each reviewer’s comment. The critical comments and useful suggestions have helped us to improve our paper considerably. As indicated in the replies that follow, we have taken these comments and suggestions into account in the revised version of our manuscript and changes indicates in ‘Track Changed’ format in the revised manuscript. The responses for review comments are attached here for your perusal.

Round 2

Reviewer 1 Report

SUMMARY

The authors of the manuscript “Mangiferin enriched biphasic calcium phosphate (containing 5% and 10% Mn doped hydroxyapatite coupled with β-TCP) favors higher Mn containing scaffolds for osteogenic and anti-bacterial efficacy” performed some modifications in the manuscript. However, relevant inconsistencies still present, according to the information below. 

MAJOR REMARKS

ABSTRACT

The abstract still has included information regarding the mineralization nodules. However, these assays were not cited in the methods, results, or discussion section. 

MATERIALS AND METHODS

Osteogenic gene expression

The authors affirmed Runt-related transcription factor X2 (RUNX2) and type-1 collagen (COL1) mRNA analysis were performed. However, the authors included results regarding osteocalcin and alkaline phosphatase gene expression. Conversely, in the abstract, the authors stated the gene expression analysis included the osteocalcin mRNA. 

Statistical analysis

The statistical analysis section is not appropriated.  

RESULTS

The authors indicated * as p>0.05 in the legend of figures 8, 10, 11, 12, 13, and 14.

In Figure 9, the authors did not indicate which scaffold was analyzed. 

Author Response

Manuscript ID: materials-2198066

Type: Article

Title: Mangiferin enriched Mn-hydroxyapatite coupled with β-TCP scaffolds exhibit osteogenicity and anti-bacterial efficacy simultaneously

Authors: Subhasmita Swain , Janardhan Reddy Koduru * , Ranjan Rautray Tapash *

Section: Advanced Nanomaterials and Nanotechnology

Special Issue: Sustainable Nanocomposites and Technologies for Water Treatment

We are very much thanking the valuable comments of editor and the all reviewers on our revised manuscript. The followings are the explanations presented in reply to each reviewers’ comment. The critical comments and useful suggestions have been helped us to improve our paper considerably. As indicated in the reply’s that follow, we have taken these comments and suggestions into account in the revised version of our manuscript and changes indicates in ‘Track Changed’ format in the revised manuscript and responses in the attached file here.

Reviewer 2 Report

No new comments.

Author Response

Thanks very much for raising good comments, which further helped improve our article.